# Clinical Features of COVID-19 Vaccine-Associated Autoimmune Hepatitis: A Systematic Review

**DOI:** 10.3390/diseases11020080

**Published:** 2023-05-30

**Authors:** Hao Zhou, Qing Ye

**Affiliations:** Department of Laboratory Medicine, Children’s Hospital, Zhejiang University School of Medicine, National Clinical Research Center for Child Health, National Children’s Regional Medical Center, Hangzhou 310000, China

**Keywords:** autoimmune hepatitis, COVID-19 vaccination, liver

## Abstract

Autoimmune hepatitis (AIH) is an inflammatory liver disease wherein the body’s immune system instigates an attack on the liver, causing inflammation and hepatic impairment. This disease usually manifests in genetically predisposed individuals and is triggered by stimuli or environments such as viral infections, environmental toxins, and drugs. The causal role of COVID-19 vaccination in AIH remains uncertain. This review of 39 cases of vaccine-related AIH indicates that female patients above the age of 50 years or those with potential AIH risk factors may be susceptible to vaccine-related AIH, and the clinical features of vaccine-associated AIH are similar to those of idiopathic AIH. These features commonly manifest in patients after the first dose of vaccination, with symptom onset typically delayed by 10–14 days. The incidence of underlying liver disease in patients with potential health conditions associated to liver disease is similar to that of patients without preexisting illnesses. Steroid administration is effective in treating vaccine-related AIH-susceptible patients, with most patients experiencing improvement in their clinical symptoms. However, care should be taken to prevent bacterial infections during drug administration. Furthermore, the possible pathogenic mechanisms of vaccine-associated AIH are discussed to offer potential ideas for vaccine development and enhancement. Although the incidence of vaccine-related AIH is rare, individuals should not be deterred from receiving the COVID-19 vaccine, as the benefits of vaccination significantly outweigh the risks.

## 1. Introduction

Coronavirus disease 2019 (COVID-19), caused by severe acute respiratory syndrome coronavirus-2 (SARS-CoV-2) infection, quickly spread worldwide in December 2019.

COVID-19 poses a significant threat to our lives and health and extensively damages society and the economy. There is an urgent need to treat and prevent COVID-19 in response to the fast-growing infection rate and escalating mortality toll, leading to COVID-19 vaccines being created and deployed at an unprecedented pace [1,2,3]. The United States Food and Drug Administration (FDA) granted an emergency use license for the Pfizer–BioNTech COVID-19 vaccine on 11 December 2020, and the Moderna vaccine on 18 December 2020. On 30 December 2020, the UK’s Medicines and Healthcare Products Regulatory Agency (MHRA) authorized the Oxford–AstraZeneca vaccine [4].

A COVID-19 vaccination was administered to at least 68% of the world’s population during the course of the following two years [5]. In phase III vaccine trials, the Pfizer–BioNTech and Moderna vaccine efficacies against COVID-19 were 91.3% and 93.2% in immunocompetent adults, respectively [6,7]. Solicited adverse events from BNT162b2 and the Moderna vaccine both involved fatigue, myalgia, arthralgia, and headache with moderate-to-severe systemic side effects. However, these effects resolved in most participants within two days and without sequelae [6,7]. Autoimmune conditions such as myocarditis and immunological thrombocytopenic purpura (ITP) are a noteworthy class of side effects associated with COVID-19 vaccinations [8,9]. There is no evidence of a causal relationship between the vaccine and autoimmune diseases due to a lack of data to investigate causality. Nevertheless, the occurrence of autoimmune diseases following COVID-19 vaccination should raise global public health concerns.

Case reports started to appear in 2021, and 39 patients with AIH-like syndromes have been reported thus far [10,11,12,13,14,15,16,17,18,19,20,21,22,23,24,25,26,27,28,29,30,31,32,33,34] (see Appendix A). Autoimmune hepatitis, described by Waldenström in 1951, is a progressive inflammatory liver disease that may lead to cirrhosis, hepatocellular carcinoma, liver transplantation, and even death [35]. AIH is distinguished serologically by high alanine aminotransferase (ALT), aspartate aminotransferase (AST), immunoglobulin G (IgG), and autoantibody positivity, as well as histologically by interface hepatitis and lymphocytic infiltration of the liver [36,37]. AIH is not limited to any one racial or ethnic group, has a female-to-male ratio of 3.6:1, and affects both children and adults of all ages [38]. In this review, we describe the populations that may be susceptible to COVID-19 vaccine-associated AIH, the clinical features of vaccine-associated AIHs and the timing of the onset of symptoms and suggest how to administer treatment for vaccine-associated AIH and precautions to take. It provides reference for clinical diagnosis and treatment of vaccine-associated AIH.

## 2. Methods

A study of AIH syndrome after coronavirus disease 2019 (COVID-19) vaccination was performed. First, on 1 October 2022, we carried out a comprehensive search of the literature in the PubMed, Embase, and Web of Science databases. The search medical subjects heading (MeSH) terms included “COVID-19”, “SARS-CoV-2”, “autoimmune”, and “hepatitis”, along with “vaccine,” “vaccination,” and “mRNA.” Second, to comprehensively collect relevant articles and cases, we eliminated all duplicate reports. Then, we screened the remaining articles based on their titles and abstracts and removed any reports that did not contain case studies, were irrelevant, or were missing critical clinical information. The filtering process is shown in Figure 1. This review was not registered in any registry and has no registration number.

The patient’s baseline characteristics were collected, such as age, sex, medication, and medical history, and clinical symptoms following the COVID-19 vaccine injection. We also gathered information on the COVID-19 vaccine, such as the type of vaccine, the timing of the first vaccination for patients, and the symptoms that appeared after the vaccine injection. Regarding AIH, we collected transaminase laboratory peak values, liver biopsies, and autoimmune antibodies. In addition, these patients’ therapeutic drugs and clinical outcomes were documented.

## 3. Characteristics of Patients with Vaccine-Associated AIH before Vaccination

A total of 39 cases of COVID-19 vaccine-associated AIH syndromes were collected from databases [10,11,12,13,14,15,16,17,18,19,20,21,22,23,24,25,26,27,28,29,30,31,32,33,34]. The median age of diagnosis was 59.0 years, with 24 (61.5%) patients over 50 years old. The oldest and youngest ages were 80 and 23, respectively, and 30 (76.9%) patients were female (Figure 2A,B). In our study, except for five patients who were not recorded clearly, nine patients had a history of liver disease, eight patients had a history of autoimmune disease, four patients had a history of medication that induced AIH, and one patient had a history of both medication and autoimmune disease (Figure 2C). This generated 22 (64.7%) patients with a history of autoimmune disease, liver disease, or medications.

Twelve patients had no history of autoimmune or hepatic disease (Figure 2C). One patient was diagnosed with gestational hypertension during pregnancy and started on labetalol 100 mg bid. Three months postpartum, she developed AIH after COVID-19 vaccination [12]. One patient had a history of hypertension treated with olmesartan and laser eye surgery two weeks prior that required topical fluoroquinolone eye drops, 1 g acetaminophen TDS, and 400 mg ibuprofen TDS for one week total [16]. One patient was on a long-term course of omeprazole, losartan, and bromazepam treatment [27]. One patient had no side effects during replacement hormone therapy after a history of early ovarian failure [29]. One patient with sarcoidosis had not received treatment [32]. Another patient received ceftriaxone due to a urinary tract infection [33]. Two patients had taken Tylenol [33]. The remaining four patients had no medication or medical history [23,26,30,31].

Upon analyzing these 39 cases, the data suggest that a majority of the vaccinated patients who experienced onset symptoms were female, constituting 30 (76.9%) of the total cases. Furthermore, the age range for a significant portion of these patients was over 50 years, representing 24 (61.5%) of the observed cases. Additionally, an intriguing observation that emerged from the analysis was that 22 (64.7%) of the patients exhibited autoimmune disorders, had a history of liver disease, or had been administered medications that have the potential to induce AIH. Thus, it can be inferred that individuals belonging to this subset of the vaccinated population should be subjected to more vigilant monitoring.

## 4. Characteristics of Patients with Vaccine-Associated AIH after Vaccination

### 4.1. Time of Onset of Symptoms

AIH was documented following the Pfizer–BioNTech, Moderna, and Oxford–AstraZeneca vaccines in 18, 15, and 6 cases, respectively. Twenty-nine patients had clinical symptoms after receiving the initial vaccination, seven patients after receiving the second vaccination, three patients after receiving both the initial and second vaccination, and one patient had no record (Table 1). This suggests that most patients develop symptoms after receiving the first dose of the vaccine. After receiving the COVID-19 vaccine, the cohort included 39 cases from our collection, and the median time to symptom onset in patients was 14.0 days. This delay period ranged from 1 to 53 days. The presentation occurred 2–53 days after vaccination against SARS-CoV-2 with Pfizer–BioNTech, with a median of 10 days. The median time from the Moderna vaccine to presentation was 12.5 days, and patients presented between 3 and 46 days. In the reported case of vaccination AstraZeneca, the onset symptom occurred at median of 18.0 days with a range of 12 to 26 days. Next, we analyzed the time of onset of the first symptoms in patients with underlying health or medication conditions and patients without preexisting conditions. The median time for the onset of symptoms after vaccination was 13 days for patients with poor health or medication status related to liver disease and the median 10 days for patients without preexisting conditions. Overall, the data suggest a 10–14-day delay in the onset of symptoms after vaccination, which is similar between the patients with underlying health related to liver diseases and the cohort with no preexisting conditions.

### 4.2. Clinical Presentation of Patients with Vaccine-Associated AIH

Twenty-eight patients clinical symptoms were documented, and 11 patients were not recorded. The most common symptoms were jaundice in 21 (75.0%) patients and fatigue in 7 (25.0%). Choluria, anorexia, pruritus, abdominal pain, and fever each accounted for five (17.9%) patients. Diarrhea and asymptomatic symptoms were observed in only one (3.6%) patient (Table 1). Idiopathic AIH may present with one or more nonspecific symptoms, and fatigue is a more common presentation [36]. However, the primary clinical manifestation of vaccine-associated AIH is jaundice and multiple nonspecific symptoms, suggesting the need for concern about liver function in people who develop jaundice from vaccination.

Interestingly, three patients among all cases experienced a recurrence of symptoms after each of the vaccinations. The first patient exhibited malaise and jaundice after receiving the first vaccine. However, as his jaundice faded and liver function tests improved, these symptoms returned shortly after the second dose [17]. One day after receiving the first vaccination dosage, the second patient initially developed nausea, vomiting, and abdominal pain. Her symptoms resolved without a therapeutic record. After receiving her boost dose of vaccine, her symptoms—choluria and jaundice—returned with greater intensity [23]. Similarly, another patient who received the first and second doses of the vaccine presented pruritus and jaundice, respectively [29]. These data suggest that this may be the onset of vaccine-induced AIH.

### 4.3. Serological Profile of Patients with Vaccine-Associated AIH

The typical biochemical features of idiopathic AIH are aspartate aminotransferase (AST) and alanine aminotransferase (ALT), ranging from slightly above the upper limit of normal to more than 50-fold, and gamma-glutamyl transferase (GGT) and alkaline phosphatase (ALP), which are usually normal or only moderately elevated [39,40]. Hepatocellular liver damage was the most prevalent pattern in our data, with transaminase levels noticeably increasing to levels close to thousands. The median ALT level was 1043.0 U/L, the median AST was 854.6 U/L, the median ALP was 193.0 U/L, the median total bilirubin level was 3.9 mg/dl, and the GGT level was 361.0 U/L (Table 1). In addition, total immunoglobulin G (IgG) levels were recorded in 24 individuals, among whom 19 (79.2%) had elevated levels. Previous work has revealed that approximately 85% of individuals with idiopathic AIH have high IgG levels [40]. These results suggest that vaccine-associated AIH may be consistent with the biochemical profile of idiopathic AIH.

Autoantibodies are a defining feature of AIH and play a significant role in the diagnostic process. Of the data collected, autoantibodies were recorded in 32 cases and were undocumented in 7. Twenty-eight patients (87.5%) tested positive for at least one of the autoantibodies. Antinuclear antibody (ANA) and anti-smooth muscle (ASMA) were positive in 25 (78.1%) and 12 (37.5%) patients, respectively, and anti-double-stranded DNA (ds-DNA) antibodies were found in 2 (6.3%) patients. Furthermore, antibodies against soluble liver antigens (anti-SLA), anti-liver cytosol type 1 (anti-LC1), and anti-neutrophil cytoplasmic antibodies (ANCAs) were detected in only one (3.1%) patient each. In contrast, autoantibody screening yielded entirely negative results in four (12.5%) patients (Table 1). In a study of idiopathic AIH, 1152 patients (88%) were positive for ANA, and 1089 patients (83%) were positive for SMA at the time of diagnosis [41]. This result is higher than the autoantibodies expressed by vaccine-associated AIH. However, some patients with vaccine-associated AIH without elevated IgG or negative autoantibodies have also been observed. Such patients should not be ignored, as elevated serum IgG and antinuclear antibody levels are not observed in some patients with acute AIH [42]. It is essential to keep in mind the possibility of acute AIH, as a delay in diagnosis and initiation of treatment can lead to a poor prognosis of AIH.

### 4.4. Liver Histology with Vaccine-Associated AIH

Histological features of liver biopsy are considered a prerequisite for diagnosing AIH [43]. The typical hallmarks of AIH are interface hepatitis with dense plasma cell-rich lymphoplasmacytic infiltrates, hepatocellular rosette formation, emperipolesis, hepatocyte swelling, and/or pyknotic necrosis [43,44]. Typically, plasma cells are plentiful at the interface and throughout the lobule, but in 34% of instances, the lack of plasma cells in the inflammatory infiltrate does not rule out the diagnosis [45].

In the cohort, liver biopsy of all patients was performed. Twenty-three (59.0%) patients showed interface hepatitis. Fourteen (35.9%) patients had centrilobular necrosis. Thirty-five (89.7%) patients showed lymphocyte or plasma cell infiltration. Another two patients were compatible with AIH but did not describe the details of live biopsy (Table 1). Twenty patients were diagnosed with AIH using the Simplified AIH score [46] or the revised original score [47]. In 19 cases, clinical symptoms, laboratory data, and liver biopsy histology supported the probable diagnosis of AIH. Moreover, eosinophils were also found in liver biopsies of 11 patients (30.8%), suggesting that liver injury may be due to drugs or toxic substances.

## 5. Treatment and Outcomes in Patients with Vaccine-Associated AIH

The aim of treatment in idiopathic AIH is to obtain complete histological and biochemical remission [48,49]. Prednisolone, prednisone coupled with or without azathioprine, or budesonide and azathioprine alternately are therapeutic medicines with a high incidence of remission and a favorable prognosis [49,50]. In all cases, steroids were used as a first-line agent in 35 (89.7%) patients (Figure 3A). One patient’s liver function test did not improve after receiving N-acetyl cysteine as a first-line therapy, and methylprednisolone had to be administered as a second medication [23]. Thirteen patients received prednisone, 14 received prednisolone, 4 received nonspecific steroids, and 1 received budesonide from the patients who received steroids as first-line therapy. Each of the remaining patients received ursodeoxycholic acid, endoscopic biliary dilation, or no treatment. All patients showed improved liver function except four who died (Figure 3B).

Four deaths in patients aged over 60 years were observed in this study. The first patient without autoimmune diseases was a 68-year-old woman who developed severe AIH after the AstraZeneca vaccination. After treatment with steroids for four weeks (1 mg/kg), the patient did not improve and developed hepatic encephalopathy and liver failure. After three days, the patient died of hepatic failure and sepsis [22]. In the second case, a 72-year-old woman was initially immunized with two doses of the AstraZeneca vaccine without incident. She was not known to have any medical conditions or be taking any medications. She was diagnosed with AIH and initiated prednisolone 40 mg once daily for two weeks, with tapering doses subsequently. Although liver function improved initially, the patient died of severe sepsis two weeks later [23]. A 77-year-old woman without autoimmune disorders presented symptoms two days after receiving the second dose of the Pfizer–BioNTech vaccine and was hospitalized the following day. After three weeks on prednisone 60 mg/day, liver tests improved significantly. Two months later, azathioprine was added but discontinued due to rash, followed by budesonide 9 mg/day instead of prednisone. Five months later, the liver enzymes were in the normal range. Unfortunately, the patient developed a possible infectious brain injury and died one month later [27]. Another patient, a 62-year-old diabetic male, had been vaccinated with AstraZeneca and, after 13 days, developed symptoms lasting three days. After treatment with 30 mg/day prednisolone, a transient improvement in liver enzymes was observed. Due to cholestasis, he also had five rounds of therapeutic plasma exchange; however, his condition did not improve. Based on the clinical presentation, the patient required liver transplantation, and due to financial constraints, the patient eventually died [31].

Among the cases in our collection, one case report demonstrated remission in a patient with vaccine-associated AIH without any treatment [33], and this phenomenon does not seem to be coincidental. A study on uncontrolled idiopathic AIH revealed that untreated asymptomatic patients had similar survival rates compared to patients undergoing immunosuppressive therapy with glucocorticoids or azathioprine [51], and spontaneous improvement of AIH may occur [52]. Therefore, this phenomenon may be justified. Should patients like this be ignored? The answer should be no. Studies indicate that a proportion of asymptomatic patients who exhibit symptoms during disease follow-up are at risk of developing end-stage liver disease with liver failure and developing hepatocellular carcinoma (HCC) [51,53]. In addition, since nearly half of all instances of idiopathic AIH have an asymptomatic subclinical course [41], it is unclear whether the cases reported thus far are the full scope of vaccine-associated AIH. However, there are no data on the overall recurrence rates and long-term outcomes for vaccine-related AIH. Therefore, we must remain vigilant and require regular follow-up and liver function testing after vaccination in patients with risk factors.

By assessing the clinical prognosis of patients diagnosed with vaccine-induced AIH, we found that treatment with steroids for vaccine-associated AIH proved effective, leading to a favorable prognosis in 35 (89.7%) patients. Regrettably, four patients were unable to recover and passed away. The treatment they received did not differ significantly from that of most patients with vaccine-associated AIH, but their deaths were attributed to sepsis, which resulted from personal variations. This highlights the importance of careful dosing and medication regimens for patients with vaccine-associated AIH to address any potential treatment-related infections. Therefore, if a bacterial infection is suspected, steroid therapy should be promptly discontinued and antimicrobial therapy initiated.

## 6. The Potential Mechanism of the Pathogenesis of AIH

While there is currently no concrete evidence of a causal link between the COVID-19 vaccine and AIH, the data we have collected suggest that a possible correlation between vaccination and the occurrence of AIH.

It is plausible that a complex interaction between vaccine components and the vaccinated individual’s susceptibility may be responsible for triggering AIH in response to COVID-19 vaccination. One potential mechanism involved in the development of vaccine-related autoimmune disorders is molecular mimicry, which refers to the similarities between peptide sequences in vaccines and those found in the human body’s self-peptides [54]. This similarity may be sufficient to lead to immune cross-interactions in which the immune system’s response to pathogenic antigens may impair similar human proteins and thus induce autoimmune diseases. The study reported similarities between the small hepatitis B surface antigen (sHBsAg) contained in the vaccine and the MS autoantigens myelin basic protein (MBP) and myelin oligodendrocyte glycoprotein (MOG)—which can be used as targets for immune cross-reaction—by comparing serum samples from 58 adults before and after receiving the HBV vaccine [55,56]. By detecting homology between HPV viral peptides and human proteins, another study indicated a significant overlap between viral and various potential SLE-associated peptides [55,56]. The autoimmune/inflammatory syndrome-induced adjuvant (ASIA) has garnered attention in recent years. Adjuvants are compounds used in the manufacture of vaccines and are designed to enhance the ability of the vaccine to produce an immune response. There are reports suggesting that adjuvants act as ligands for Toll-like receptors (TLRs) through molecular mimicry, triggering the activation of the TLR pathway and the production of type I interferons (IFNs) and proinflammatory cytokines. Moreover, adjuvants activate dendritic cell recruitment through chemotaxis and antigen presentation, resulting in a more robust B cell and T cell response. Ultimately, this leads to an increased adaptive immune response against the antigen [57,58]. To date, the pathogenic mechanism of COVID-19 vaccine-associated AIH is not known. However, recent research indicated that SARS-CoV-2 antibodies produced moderate-to-strong responses with 21 out of 50 tissue antigens [59], suggesting that molecular mimicry likely plays a role. Molecular mimicry results in the production of homologous self-antigens [60], leading to autoimmune diseases [36].

The hypothesized molecular mechanism of autoimmune-mediated liver injury is shown in Figure 4. The presentation of a self-antigenic peptide to the T cell receptor (TCR) of T helper cells (Th0) by antigen-presenting cells (APCs). In healthy conditions, T cells that recognize self-antigens undergo apoptosis by clonal deletion or differentiate into anergic T cells. TGF-β stimulates the differentiation of Th0 cells into regulatory T cells (Tregs), which have immunosuppressive effects [61]. However, in abnormal immune conditions, immune cells against self-antigens remain active and cause autoimmune diseases. The uncommitted Th0 cells differentiate into Th1, Th2, or Th17 cells depending on the cytokine environment [62,63,64]. Th1 cells secrete interleukin-2 (IL-2) and interferon-γ (IFNγ), stimulating CD8+ cells to recognize the antigen–major histocompatibility complex (MHC) class I complex and activating macrophages to secrete IL-1 and tumor necrosis factor (TNFα) to recognize MCH II, respectively. The Th1 cell proportion and IFN-γ secretion were lower in healthy controls than in patients with AIH [39,63,64]. Th2 cells, on the other hand, secrete cytokines (e.g., IL-4, IL-10, and IL-13) that may stimulate self-antigen reactive B cells that produce autoantibodies [63,64]. Autoantibodies target liver cells and induce damage through natural killer (NK) cells and complement-mediated cytotoxicity. Th17 cells produce proinflammatory cytokines (e.g., IL-17 and IL-22) and TNF-α, leading to the induction of hepatic secretion of IL-6. While the existence of Th17 cells in AIH has been reported, their precise role in disease pathogenesis remains incompletely understood [62,64].

## 7. Limitations

Our study has several limitations that must be acknowledged. First, limited records on the medication and health history of the patients prior to vaccination could have led to oversights of potential risk factors predisposing to AIH. Additionally, incomplete documentation of patient symptoms and laboratory values in the original article resulted in a limited diagnosis of immune-mediated hepatitis following vaccination. Second, due to the possibility of publication bias in case reports and partial asymptomatic patients, it is difficult to accurately determine the incidence of vaccine-related AIH. This could potentially lead to underestimation of the risks involved. Third, as this manuscript lacked data that could be available to examine causal relationships, for example, lack of control data and insufficient number of cases; it is not possible to establish a causal relationship between the vaccine and AIH with the current data.

## 8. Conclusions

In these 39 cases, the majority of patients were women, over 50 years old, and with potential AIH risk factors, and there are no available data to date to check that AIH occurs after vaccination. Furthermore, vaccine-associated AIH seems to present with consistent clinical features, including clinical symptoms, biochemical features, autoantibodies, and liver biopsy findings, compared to idiopathic AIH. Patients who are asymptomatic after vaccination should be closely monitored, and their liver function should be evaluated. Moreover, while steroid therapy is effective in treating vaccine-associated AIH, it is noteworthy that individual variability may lead to sepsis caused by bacterial infection. Finally, these risk factors should not deter individuals from receiving the COVID-19 vaccine, as a physician has a duty to promote vaccination while being cognizant of potential risks and striving to enhance current medical practice and minimize harm.

## Figures and Tables

**Figure 1 diseases-11-00080-f001:**
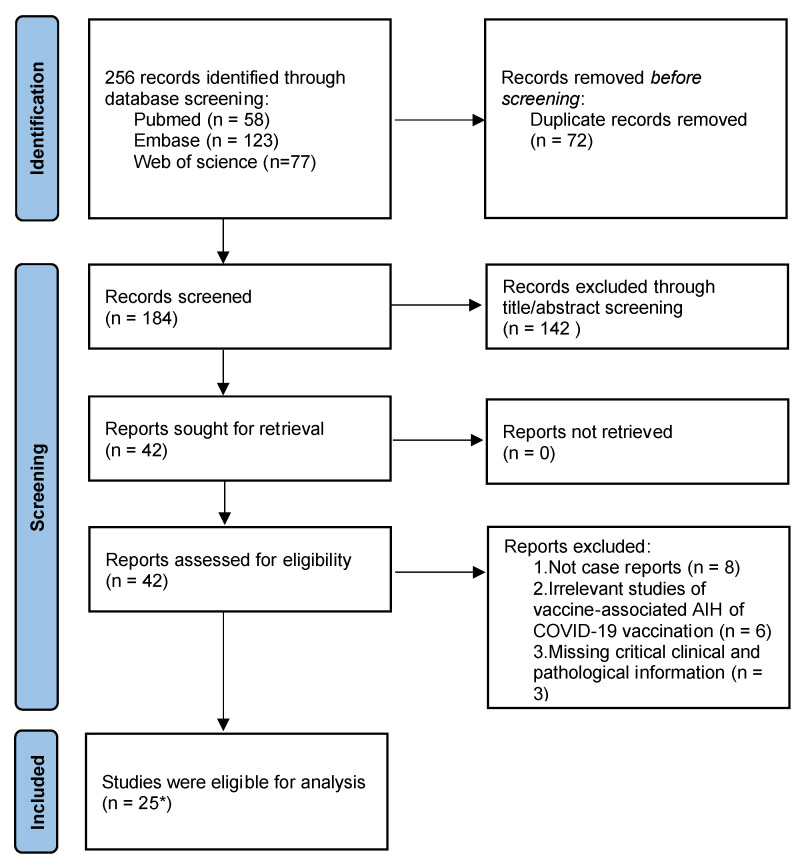
PRISMA flowchart describing the process of selecting eligible studies. * of the 25 articles included in the review, two contained three cases, two contained two cases, one was a series of case reports with nine cases, and the rest consisted of a single case. AIH, autoimmune hepatitis.

**Figure 2 diseases-11-00080-f002:**
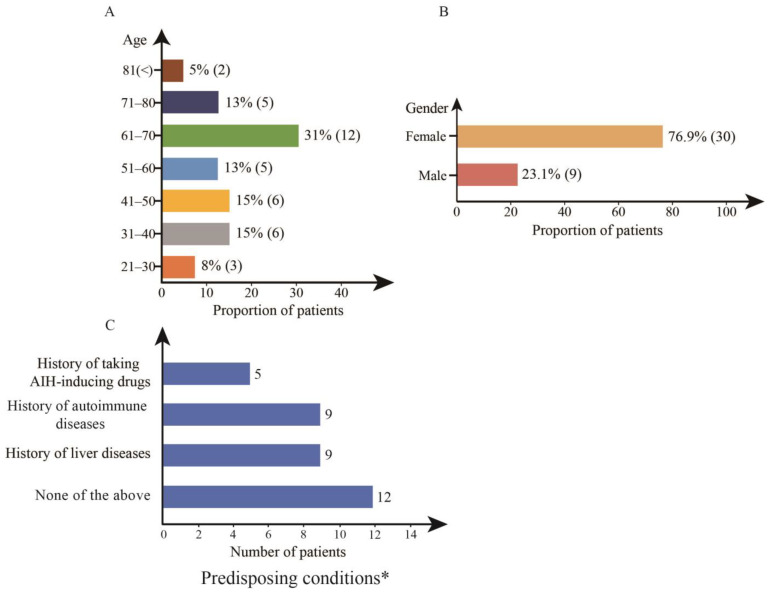
Pre-vaccination characteristics of patients with vaccine-associated AIH. Patient age (**A**), sex (**B**), and predisposing conditions (**C**). * one patient had a history of both taking AIH-inducing drugs and autoimmune diseases.

**Figure 3 diseases-11-00080-f003:**
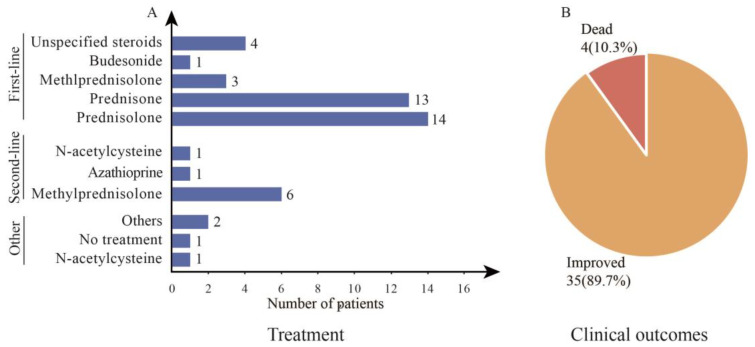
Treatment and outcomes of patients with vaccine-associated AIH. Patient medication (**A**) and clinical outcomes (**B**).

**Figure 4 diseases-11-00080-f004:**
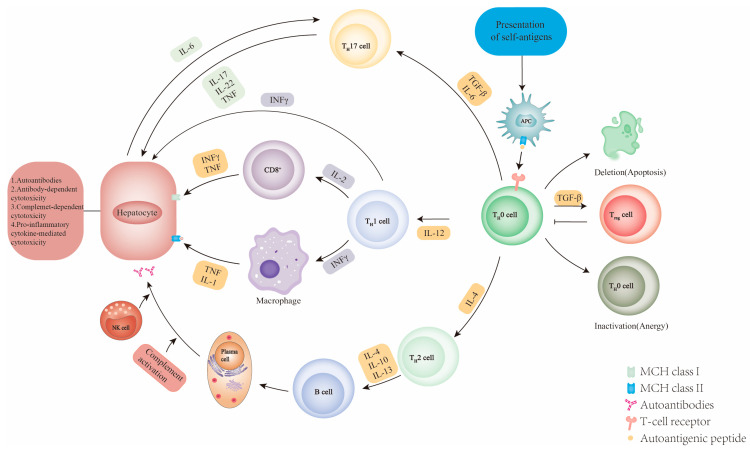
The potential mechanism of the pathogenesis of autoimmune hepatitis. APC, antigen-presenting cell; IFNγ, interferon-γ; MHC, major histocompatibility complex; NK, natural killer; TCR, T cell receptor; TGFβ, transforming growth factor-β; T_H_0, naive CD4+ T helper; T_H_1, T helper 1; T_H_2, T helper 2; T_H_17, T helper 17; TNF, tumor necrosis factor; Treg, regulatory T.

**Table 1 diseases-11-00080-t001:** Clinical presentation of vaccine-associated AIH.

Type of Vaccines ^a^	No. of Patients (%)	Autoantibodies ^d^	No. of Patients (%)
Pfizer–BioNTech	18 (46.2%)	ANA	25 (78.1%)
Moderna	15 (38.4%)	ASMA	12 (37.5%)
Oxford–AstraZeneca	6 (15.4%)	ds-DNA	2 (6.3%)
**Time of onset of symptoms ^b^**	No. of Patients (%)	Anti-SLA	1 (3.1%)
First vaccination	31 (81.6%)	Anti-LC1	1 (3.1%)
Second vaccination	7 (18.4%)	ANCA	1 (3.1%)
**Symptoms ^c^**	No. of Patients (%)	None	4 (12.5%)
Jaundice	21 (75.0%)	**Serum biochemical** **parameters ^e^**	Median (Range)
Fatigue	7 (25.0%)	ALT(U/L)	1038 (171–2664)
Anorexia	5 (17.9%)	AST (U/L)	862 (111–2314)
Choluria	5 (17.9%)	ALP (U/L)	186 (24–2252)
Anorexia	5 (17.9%)	Tbil (U/L)	3.84 (0.33–45)
Pruritus	5 (17.9%)	GGT (U/L)	345 (98–810)
Abdominal pain	5 (17.9%)	Total IgG (mg/dL)	1998 (1081–4260)
Fever	5 (17.9%)	Liver biopsy ^a^	No. of Patients (%)
Diarrhea	1 (3.6%)	Interface hepatitis	23 (59.0%)
Asymptomatic	1 (3.6%)	pycnotic necrosis	14 (35.9%)
		Lymphocyte/plasma cells infiltration	35 (89.7%)
		Eosinophils	11 (30.8%)

^a^: Thirty-nine patients had recorded the type of vaccine and liver biopsy. ^b^: One out of 39 patients had no record of the type of vaccine. ^c^: Eleven out of 39 patients had no documentation of symptoms. ^d^: Seven out of 39 patients had no record of autoantibodies. ^e^: The result of ALT, AST, ALP, Tbil, GGT, and total IgG were recorded for 39, 28, 28, 35, 13, and 26 patients, respectively. ALT, alanine aminotransferase; AST, aspartate aminotransferase; ALP, alkaline phosphatase; TBL, total bilirubin; GGT, gamma glutamyl transferase; IgG, immunoglobulin G; ANA, anti-nuclear antibodies; ASMA, anti-smooth muscle antibodies; ds-DNA; anti-double stranded DNA; anti-SLA, antibodies against soluble liver antigen; anti-LC1, anti-liver cytosol type 1.

## Data Availability

Not applicable.

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
