# Peer review of "Clinical Features of COVID-19 Vaccine-Associated Autoimmune Hepatitis: A Systematic Review"

_diseases, 2023, doi:10.3390/diseases11020080_

Round 1

Reviewer 1 Report

please see the attachment file.

Reviewer 2 Report

Dear Authors.

Thank you for the opportunity to review the article "Clinical Features of COVID-19 Vaccine-Associated Autoimuune Hepatitis: a narrative review (disease-2346385)." This manuscript presents a well-structured literature review on the relationship between the COVID-19 vaccine and AIH. Nonetheless, there are a few concerns (#1 to #8) that warrant attention. Kindly review these points meticulously and make the necessary amendments. It is advised that special attention be given to addressing the inaccuracies in the portrayal of causality, along with enhancing the descriptions of the study's novelty, significance, and knowledge gaps. This would contribute to the overall quality and clarity of the manuscript.

Best regards,

Comments

#1. Background

The information in the Background section was related to the COVID-19 epidemic, vaccines, and autoimmune diseases (myocarditis, ITP, AIH). The authors need to add this study's novelty, importance, and knowledge gap to show readers why and how this study is essential. Please consider adding this information.

#2. Figure 1

Figure 1 shows the final number of included studies is "25". However, the number of cases included is 39. This difference is because more than one case was included in one study. This is aleardy written in the supplemental material. However, not all readers will read the supplementary material. If the authors do not explain this difference in the text or a footnote of the figure, readers may think it is a misstatement. Please consider adding this in the text or a footnote.

#3. Figures 2 (A) and (B)

Pie charts are not suitable for reading proportions. For example, it is not easy to decide on which is larger at a glance between "41-50" and "51-60" in Figure 2 (A). If there is a numerical description (i.e., 6 (15%) and 5 (13%)) as in the manuscript, it is possible to tell which is larger, but it isn't easy to do so with only a figure. A figure that does not allow for correct comparison of sizes without looking at the numbers is a real downfall. I suggest replacing the pie chart with a bar chart. A bar chart allows one to accurately determine the relationship between the size of each category at a glance. Please give favorable consideration to this change.

#4. Figure 2 (C)

In the text (Line 104), it is stated that "22 (64.7%) of the patients." However, in Figure 2 (C), the total number of patients is 23 (= 5 + 9 + 9). I presume this difference in numbers is because one patient is counted in more than one category. I ask that the authors carefully explain these numerical differences in the text or a footnote of the figure. Such a statement will help avoid readers' misinterpretation and enhance the study's value. Please consider this modification.

#5. Table 1

Table 1 lists percentages, but several appear to be incorrect. For example, Diarrhea is listed as "1 (5.0%)"; symptoms were not documented in 11 patients, so the denominator is 28 (= 39 - 11). Therefore, 1/28 ≈ 0.0357 ≈ 3.6%. There seem to be other errors like this one, so please inspect carefully and list the correct value. Also, please clearly indicate the total number (i.e., the denominator) for each item in the table or a footnote. Is the total number of each item correct as follows?

Type of vaccines: 38 (=39 - 1)

Time of onset of symptoms: 38 (=39 - 1)

Symptoms: 28 (=39-11)

Autoantibodies: 39

Serum biochemical parameters: 39

Liver biopsy: 39

#6.Causality

In the text (Lines 264 to 265), there is a reference to "the data we have ... a coincidence following vaccination." This statement suggests a causal relationship (or at least mentions a correlation) between vaccination and AIH. However, this study only included vaccinated AIH patients. To mention a causal relationship or correlation, it is needed data on patients (a) to (d) listed below. Regrettably, only (a) is applicable to this study, and it would be inappropriate to mention causal relationships. Naturally, this extends to correlations as well.

(a) Vaccinated AIH patients

(b) Vaccinated non-AIH patients

(c) Non-vaccinated AIH patients

(d) Non-vaccinated non-AIH patients

#7. Limitation

In the text (Lines 313 to 315), there is a reference to "Third, as all the ... our analysis of cases".  I agree with you that causality cannot be mentioned in this study. However, I am skeptical that the reason for this is that it is an observational study. As per comment #6 above, I believe that this is due to the lack of data available to examine causal relationships. It is possible to refer to a causal relationship if the observational study was carefully designed and data were obtained (For example, the occurrence of neonatal malformations due to thalidomide and lung cancer due to smoking have been mentioned as causal relationships in observational studies). The rationale for the limitation warrants revision. Kindly give this suggested modification thoughtful consideration.

#8. Conclusion

In the text (Lines 319 to 321), the phrase "Based on the analysis ... risk factors for AIH" requires revision, as the assertion of causality may be misleading to readers. With reference to (a) in #6 above, it is important to recognize that even if an individual is over 50 years old, female, and possesses multiple existing risk factors, these factors do not necessarily increase the likelihood of developing AIH after vaccination. For instance, if the authors were to gather data for (a), (b), (c), and (d), then for (a) and (b) in #6 above, most of the study participants might be over 50 years old, female, and have numerous pre-existing risk factors. In such a case, these attributes dose not be associated with the development of AIH. As a result, it is recommended that this section of the statement be thoughtfully modified.

There appear to be no concerns regarding the quality of the English language used in the manuscript.

Reviewer 3 Report

Thank you for sharing the lengthy and comprehensive manuscript. Here just some suggested edits that could help to improve the article:

L47-49: The content of the sentence isn't fully clear, consider revising.

L63-65: Did you apply any restrictions in terms of article type, e.g., only case reports? 

L122/123/125/127: Did you generate the median or mean time of symptom onset? Please clarify in your manuscript.

L161/163: Against what are the IgG levels directed? 

Table 1: Regarding ALT, AST, ALP, TbiI, GGT and serum IgG, how many decimal places have the ranges reported? 

L320: Revise "over 50" to "over 50 years".

NA

Round 2

Reviewer 2 Report

I have now reviewed your revised manuscript, and I would like to express my sincere gratitude for your efforts. It is clear to me that you've thoughtfully addressed my previous comments and made appropriate revisions.

While the study does indeed have certain limitations due to data constraints, I would like to express my commendation for conducting this scientific investigation to the utmost of your abilities, given the constraints you faced.

At this juncture, I find myself without further suggestions or comments to add. Thank you once again for the opportunity to engage with your work.

I have no comments about the quality of English.